# Zero- to One-Dimensional Zn_24_ Supraclusters: Synthesis, Structures and Detection Wavelength

**DOI:** 10.3390/nano13233058

**Published:** 2023-11-30

**Authors:** Yating Chen, Zhonghang Chen, Jiming Wang, Xuandi Ma, Linyu Yuan, Shuhua Zhang, Fushun Tang

**Affiliations:** 1Guangxi Key Laboratory of Electrochemical and Magnetochemical Functional Materials, College of Materials Science and Engineering, Guilin University of Technology, Guilin 541004, China; 1020180460@glut.edu.cn (Y.C.); 1020180495@glut.edu.cn (X.M.);; 2College of Chemistry, Guangdong University of Petrochemical Technology, Maoming 525000, China

**Keywords:** supracluster, 24 nuclear, fluorescence quenching, detect wavelengths, limit of detection

## Abstract

A zinc supracluster [Zn_24_(ATZ)_18_(AcO)_30_(H_2_O)_1.5_]·(H_2_O)_3.5_ (**Zn_24_**), and a 1D zinc supracluster chain {[Zn_24_(ATZ)_18_(AcO)_30_(C_2_H_5_OH)_2_(H_2_O)_3_]·(H_2_O)_2.5_}_n_ (**1-D⊂Zn_24_**) with molecular diameters of 2 nm were synthesized under regulatory solvothermal conditions or the micro bottle method. In an N,N-dimethylformamide solution of **Zn_24_**, Fe^3+^, Ni^2+^, Cu^2+^, Cr^2+^ and Co^2+^ ions exhibited fluorescence-quenching effects, while the rare earth ions Ce^3+^, Dy^3+^, Er^3+^, Eu^3+^, Gd^3+^, Ho^3+^, La^3+^, Nd^3+^, Sm^3+^, and Tb^3+^showed no obvious fluorescence quenching. In ethanol solution, the **Zn_24_** supracluster can be used to selectively detect Ce^3+^ ions with excellent efficiency (limit of detection (LOD) = 8.51 × 10^−7^ mol/L). The **Zn_24_** supracluster can also detect wavelengths between 302 and 332 nm using the intensity of the emitted light.

## 1. Introduction

In recent years, polynuclear metal complexes have received considerable attention due to their functional applications in science and technology, exhibiting magnetic [1,2,3,4], fluorescence [5,6,7,8], optical [9,10,11,12], electronic [13], optoelectronic [14] and catalytic properties [15,16,17]. In addition, they are used in the treatment and diagnosis of various diseases [18,19], as well as in the components of sensors [20,21,22]. Among the diverse transition-metal polynuclear complexes, zinc complexes show unique properties and variable structural fluorescence [22] and catalytic [15,17] properties. These properties are due to the d^10^ electron shell structure of Zinc(II) that has ideal flexible coordination modes and various coordination numbers. In addition, many aspects of the synthesis affect the structure and nuclear number of transition-metal polynuclear complexes, including the metal ions, ligands, concentrations, counter-ions, templates, solvents, temperatures, and pH [23,24,25].

The careful selection of an appropriate organic ligand with specific characteristics, such as variable bonding modes and the ability to engage in supramolecular interactions, can facilitate the tailoring and construction of clusters with desirable properties [26,27,28]. Based on the advantages of abundant coordination modes (Figure 1), multiple coordination sites, strong binding ability, and a particular orientation [29], tetrazole is desirable for the preparation of metal cluster/cage coordination polymers (CPs) with rich node–linker connectivity, diverse one- through three-dimensionality, specific topological structures, and superior physicochemical properties [30,31,32,33]. Among these compounds, nonadecanuclear sliver cluster-based CPs have the highest nuclear number of tetrazolium and its derivatives [34]. Using 5-amino-1,2,3,4-tetrazole (Hatz), we synthesized a zinc supracluster [Zn_24_(ATZ)_18_(AcO)_30_(H_2_O)_1.5_]·(H_2_O)_3.5_ (**Zn_24_**), and a zinc supracluster chain {[Zn_24_(ATZ)_18_(AcO)_30_(C_2_H_5_OH)_2_(H_2_O)_3_]·(H_2_O)_3_}_n_ (**1-D⸦Zn_24_**). To the best of our knowledge, both **Zn_24_** and **1-D⸦Zn_24_** are the largest cluster or cluster-based CPs constructed using tetrazole and its derivatives. In particular, the **Zn_24_** supracluster can detect wavelengths of light in the range of 300–340 nm.

## 2. Experimental Methodology

### 2.1. Materials and Physical Measurements

All chemicals were bought commercially and used directly after receipt. Elemental analyses were performed using a Perkin-Elmer 240 elemental analyzer (CHN). The FT-IR spectra were captured in the 4000–400 cm^−1^ region from KBr pellets on a Bio-Rad FTS-7 spectrometer. The SHELXL crystallographic program for molecular structures was used to determine the X-ray crystal structures using an Agilent G8910A CCD diffractometer. Photoluminescence experiments were performed using a Hitachi F-4600 fluorescence spectrophotometer. The power X-ray diffraction (PXRD) patterns were determined using a PANalytical X’Pert^3^ power diffractometer (operating at 40 kV and 40 mA) with graphite-monochromatized Cu Kα radiation (*λ* = 1.54056 Å). ^1^ H NMR spectra were recorded on Bruker AVANCE III 500 instruments.

### 2.2. Synthesis of L^1^H_2_–L^5^H_2_

A mixture of 5-amino-1,2,3,4-tetrazole (Hatz) (10 mmol), salicylaldehyde derivatives (10 mmol), and ethanol (20 mL) was refluxed at 353 K for 1 h in a 100 mL flask. A beige precipitate of L^n^H_2_ formed, and it was then rinsed three times with fresh ethanol (10 mL × 3) and dried at 50 °C for 24 h (refer to the ESI† for details).

### 2.3. Synthesis of Zn_24_

A mixture of H_2_L^1^ (0.5 mmol, 0.1340 g), Zn(CH_3_COO)_2_·2H_2_O (0.5 mmol, 0.1048 g), and ethanol (10 mL) was stirred for 30 min, with the pH adjusted to 6 through the addition of triethylamine. The mixture was then sealed in a 15 mL Teflon-lined stainless-steel vessel, and heated at 353 K for 48 h in an oven, followed by slow cooling to room temperature. Four-cornered golden yellow crystals in a double-cone shape were collected, washed with ethanol, and dried in air. Phase-pure **Zn_24_** crystals were obtained through manual separation (yield: 66.5 mg, ca. 64.33% based on Zn(II)). *Anal. Calc*. for **Zn_24_**: C_78_H_138_N_90_O_66_Zn_24_ (*M*r = 4961.66), *calc*.: C, 18.88; H, 2.80; N, 25.39%. Found: C, 18.79; H, 2.87; N, 25.46%. The FT-IR data for **Zn_24_** (Appendix A, KBr, cm^−1^) were as follows: 3445 s, 1578 m, 1400 w, 1165 w, 1101 m, 941 w, 758 w, 685 w, 616 w, and 483 w.

### 2.4. Synthesis of 1-D⊂Zn_24_

A mixture of H_2_L^1^ (0.5 mmol, 0.1340 g), Zn(CH_3_COO)_2_·2H_2_O (0.5 mmol, 0.1048 g), ethanol (10 mL) and acetonitrile (2 mL) was stirred for 30 min with the pH adjusted to 6 through the addition of triethylamine. The mixture was then sealed in a 20-mL micro bottle capable of autonomously adjusting the reaction pressure and heated at 343 K for 48 h. Subsequently, the micro bottle was slowly cooled to room temperature. Four-cornered golden yellow crystals with a double-cone shape were collected, washed with ethanol and dried in air. Phase-pure crystals of **1-D⊂Zn_24_** were obtained through manual separation (yield: 70.2 mg, ca. 66.67% based on Zn(II)). *Anal. Calc*. for **1-D⊂Zn_24_**: C_82_H_150_N_90_O_68_Zn_24_ (*M*r = 5053.79), *calc*.: C, 19.01; H, 2.99; N, 24.93%. Found: C, 18.92; H, 3.06; N, 25.04%. The FT-IR data for **Zn_24_** (Appendix A, KBr, cm^−1^) were as follows: 3445 s, 1578 m, 1400 w, 1165 w, 1101 m, 941 w, 758 w, 685 w, 616 w, and 483 w.

### 2.5. Single-Crystal X-ray Diffraction

The single-crystal data of the **Zn_24_** and **1-D⸦Zn_24_** complexes were collected using a SuperNova (single source at offset) Eos with graphite monochromatic Mo-Kα radiation (*λ* = 0.71073 Å) in the *ω* scan mode in the ranges of 3.16° ≤ *θ* ≤ 25.01° and 3.30° ≤ *θ* ≤ 25.01°, respectively. Raw frame data were integrated using the SAINT program [35]. The **Zn_24_** and **1-D⸦Zn_24_** structures were solved with direct methods using SHELXS [35] and refined with full-matrix least-squares on *F*^2^ using SHELXL-2018 within the Olex2 GUI [36]. Empirical absorption correction using spherical harmonics was implemented in the SCALE3 ABSPACK scaling algorithm. All non-hydrogen atoms were refined anisotropically. All hydrogen atoms to carbon atoms were positioned geometrically and refined as riding atoms. Calculations and graphics were performed with *SHELXTL* [35]. The computer programs used in this study were CrysAlis PRO (Agilent Technologies, Version 1.171.37.35 released 13-08-2014 CrysAlis171.NET compiled 13 August 2014), SHELXL [35], and Olex2 [36]. The crystallographic details of **Zn_24_** and **1-D⸦Zn_24_** are provided in Table 1. Selected bond lengths and angles for **Zn_24_** and **1-D⸦Zn_24_** are listed in Appendix A.

## 3. Results and Discussion

### 3.1. Structural and Synthetic Details

Herein, we investigated the effects of ligand, reaction temperature, counterbalance anion, ligand/metal ion molar ratio, solvent, and synthetic method on the self-assembly of supraclusters (Figure 2). The synthetic strategy for the Hatz system is depicted in Figure 2. First, a mixture of Zn(OAc)_2_·2H_2_O (0.2 mmol), 4-bromo-2-[(1H-tetrazol-5-ylimino)-methyl]-phenol (**H_2_L^1^**, 0.2 mmol), and anhydrous ethanol (10 mL) was poured into a Teflon-lined autoclave (20 mL). The autoclave was cooled slowly to room temperature after heating at 80 °C for 2 days. Four-cornered golden yellow **Zn_24_** crystals in a double-cone shape were collected via filtration. In the **Zn_24_** supracluster, Hatz is produced by the decomposition of **H_2_L^1^**. To understand the role of **H_2_L^1^** in the synthesis, we used various salicylaldehyde-derived Schiff bases (**H_2_L^2^**-**H_2_L^5^**) instead of 4-bromo-2-[(1H-tetrazol-5-ylimino)-methyl]-phenol (**H_2_L^1^**) to conduct the same experiment, but we could not obtain the **Zn_24_** supercluster or analog. Similarly, if only **H_2_L^1^** was replaced by Hatz, the **Zn_24_** supercluster or analog was not obtained. Through previous experiments, we can draw the conclusion that although 5-bromosalicylicaldehyde does not participate in coordination in the 24-atom Zn cluster, 5-bromosalicylicaldehyde is an essential raw material for the synthesis of the **Zn_24_** cluster. Thus, we speculate that 5-bromosalicylicaldehyde may act as a template.

According to the ring structure of **Zn_24_**, the **Zn_24_** cluster can form a 1D chain or 2D network through bridging ligands. **H_2_L^1^** and Zn(CH_3_COO)_2_·2H_2_O were selected as the starting materials. By tuning the reaction temperature, solvent, ligand/metal salt molar ratio, and synthetic methods, the optimal synthesis conditions for the 1D Zn supracluster chain **1-D⸦Zn_24_** were determined as follows: reaction temperature, 70 °C; solvent, anhydrous ethanol (8 mL), and acetonitrile (2 mL); **H_2_L^1^**/Zn(CH_3_COO)_2_·2H_2_O molar ratio, 2:1; and reactor, micro bottle (autonomously adjusting the reaction pressure).

To obtain a 2D supracluster network, we replaced acetic acid with various carboxylic acids such as oxalic acid, malonic acid, succinic acid, and terephthalic acid. However, these experiments were unsuccessful.

### 3.2. Crystal Structures of Zn_24_ and 1-D⸦Zn_24_

Single crystal analysis confirmed **Zn_24_** to have a 0D wheel-like coordination supracluster of the monoclinic crystal system with *P*2_1_/*n* space group consisting of 24 Zn^II^ atoms, 30 acetate groups, 1.5 coordinated water molecules, 3.5 lattice water molecules and 18 Atz ligands derived from H_2_L^1^ (Figure 1a). The Zn_24_ cluster was stabilized by 5-amino-1,2,3,4-tetrazole, which binds along the wheel of the cluster core (Figure 1b), bridging the four neighboring zinc atoms. Acetate groups further stabilized the cluster through 18 *μ*_2_:*ƞ*^1^:*ƞ*^1^-acetate bridging two zinc atoms. The 12 Zn ions in the inner ring of the wheel (inner red ring shown in Figure 1a) coordinated with four N atoms from four different Atz ligands and two O atoms from two *syn-syn-μ*_2_:*ƞ*^1^:*ƞ*^1^-acetate bridging groups to form a distorted octahedral geometry. By contrast, the 12 Zn atoms in the outer ring (outer red ring shown in Figure 1a) coordinated with two N atoms from two different Atz ligands. They also coordinated with two or three O atoms from one *syn*-*syn-μ*_2_:*ƞ*^1^:*ƞ*^1^-acetate bridging group and one *μ*_1_:*ƞ*^1^:*ƞ*^1^-acetate terminal group or one *μ*_1_:*ƞ*^1^-acetate ligand, as well as one coordinated water molecule, to form a distorted tetragonal pyramidal, or a trigonal bipyramidal or an octahedral geometry. Close inspection of the nanosized wheel-like conformation showed approximate wheel dimensions of 8.912 × 20.491 × 9.747 Å (inner ring diameter × outer ring diameter × wheel thickness), where the inner ring diameter is the distance between tetrazole planes (i.e., planes (N31,N32,N33,N34,C27) and (N31, N32, N33, N34, C27)^i^, symmetry code: (i)-*x*,-*y*,-*z*); the outer ring diameter is N30····N30^i^; and the wheel thickness is the distance between the C13-C29^i^-C26^i^ -plane and the C13^i^-C29-C26-plane.

Complexes **1-D⸦Zn_24_** and **Zn_24_** have similar basic structures, i.e., the **Zn_24_** supracluster. Complex **1-D⸦Zn_24_** was constructed as a 1D Zn_24_ suprachain through the double *syn*-*anti- μ*_2_:*ƞ*^1^:*ƞ*^1^-bridging acetic group linking of Zn_24_ (Figure 2).

### 3.3. Luminescent Properties

Numerous studies have demonstrated the good fluorescence of clusters, especially those of Zn(II) and Cd(II) ions with closed d subshells [37,38]. In recent years, Zn(II) clusters have been widely used in luminescent probes due to their desirable advantages in detection and promising applications in biological and environmental systems [39]. Luminescent probes and complexes can selectively detect various sizes of molecules or ions through their adjustable porosity [40].

In this paper, the luminescent properties (the phase purity of **Zn_24_** has been checked using PXRD patterns, Appendix A) of **Zn_24_** were investigated in different solvents with concentrations of 1 × 10^−6^ mol·L^−1^ (Figure 3). Upon photoexcitation at 404, 382, 426 and 418 nm in water, N,N-dimethylformamide (DMF), DMSO, and ethanol solvent, **Zn_24_** exhibited green, blue, green and green luminescent emission bands with fluorescence maxima at 496, 459, 507, and 508 nm, respectively. These results predominantly originated from the metal-to-ligand charge-transfer excited state [41,42]. Furthermore, **Zn_24_** exhibited a qualitative change in its luminescence due to the interaction between metal ions and ligands, and it had a stronger fluorescence intensity in DMF and ethanol solutions. Although **Zn_24_** also had a stronger fluorescence intensity in DMSO, we did not consider DSMO for **Zn_24_** luminescent probes due to its high toxicity. Thus, we discussed the **Zn_24_** complex as a luminescent probe for highly selective sensing in DMF and ethanol.

**Zn_24_** (1 mg) was immersed in 10 mL of DMF solutions containing MCl*_n_* (M = Al^3+^, Ba^2+^, Co^2+^, Cr^2+^, Cu^2+^, Fe^3+^, Mn^2+^, Ni^2+^, Pb^2+^, or Zn^2+^) to form complex suspensions incorporating various metal ions for luminescence studies, and Fe^3+^, Ni^2+^, Cu^2+^, Cr^2+^, and Co^2+^ ions all demonstrated a fluorescence-quenching effect (Figure 4), indicating that **Zn_24_** was not selective toward ions in DMF solution. At the same time, the rare earth ions (Ce^3+^, Dy^3+^, Er^3+^, Eu^3+^, Gd^3+^, Ho^3+^, La^3+^, Nd^3+^, Sm^3+^, and Tb^3+^) have no clear fluorescence-quenching effects (Appendix A). However, **Zn_24_** showed high selectivity in ethanol solution. The luminescence intensity of **Zn_24_** revealed that the addition of Ce^3+^ can lead to complete quenching in ethanol solution compared with other metal ions (Figure 5).

The **Zn_24_** (1 mg, with a final concentration of 1 mg/mL in ethanol) and different concentrations of Ce^3+^ (from 1 × 10^−2^ to 1 × 10^−8^ mol/L) were added to the sample tube at room temperature. The fluorescence spectrum was taken at its excitation wavelength (*λ =* 402 nm). 

The fluorescence spectra of the Zn_24_-Ce^3+^ system for various concentrations of Ce^3+^ are shown in Figure 6. The fluorescence intensity at 504 nm progressively increased as the concentration of Ce^3+^ decreased. In addition, we quantitatively analyzed the quenching efficiency through the Stern–Volmer equation: *I*_o_/*I* = *K*_sv_[*C*] + l, where *I*_o_ and *I* are the respective emission intensities before and after adding Ce^3+^, while *C* is the concentration of Ce^3+^ in ethanol solution. The quenching efficiency, of **Zn_24_** was −1.68 × 10 M^−1^ (Appendix A). According to the limit of detection (LOD) = 3*δ*/*K*_sv_ (Appendix A), we calculated an LOD of 8.51 × 10^−7^ mol/L, which was lower than the reported LOD of the Ln-MOF [41]. 

### 3.4. Detection Wavelength

The solid-state fluorescence spectra of Hatz were obtained at a slit width of 5 nm and an excitation wavelength of 402 nm, while the solid-state fluorescence spectra of **Zn_24_** were obtained at an excitation wavelength of 302–332 nm (Figure 7 and Appendix A). Under the same test conditions, the fluorescence spectrum of Hatz peaked at 615 nm, while that of **Zn_24_** peaked at 502 nm. The luminous color changed from red to blue–green, and the fluorescence intensity of **Zn_24_** was more than 200 times that of Hatz. The full-type Zn^2+^ metal ion in **Zn_24_** has an extra-nuclear d^10^ electron, and did not undergo a d–d transition, leading to a significant enhancement in luminous intensity. At the same time, the deprotonated tetrazolium ring is an electron-deficient conjugated ring that causes the electron migration (M→L) of zinc ions to the tetrazolium ring [42,43]. As a result, the luminous color changed from red to blue–green, and the fluorescence intensity of **Zn_24_** was more than 200 times that of Hatz. The fluorescence intensity of **Zn_24_** decreased linearly with the increase in excitation wavelength (Figure 7 and Figure 8). Between 302 and 332 nm, the wavelength of the excitation light can be determined by detecting the intensity of the emitted light.

### 3.5. Hirshfeld Surface Analysis of the Complex **Zn_24_**

Hirshfeld surface analysis [44] is a useful tool for describing the surface characteristics of molecules, and was performed to visualize the different intermolecular interactions in crystal structures by employing 3D molecular surface contours. Figure 9 displays the findings of the **Zn_24_** Hirshfeld surface study. The middle shape index ranges from −1.000 to 1.000 Å, whereas the range of the d_norm_ surface on the left is −1.238 to 1.570 Å. The range of the curvature curvedness was −4.000 to 0.400 Å. In Figure 9, the d_norm_ surface map of **Zn_24_** is colored from light to dark red spots to represent the interaction force of the complex **Zn_24_** from weak to strong.

One useful supplement for Hirshfeld surface analysis is the 2-D fingerprint plot [45]. It quantitatively analyses the nature and type of intermolecular interaction between the molecules inside the crystals. The fingerprint plots can be decomposed to highlight particularly close contacts between the elements (Figure 10). The H···H interaction is one of the most significant contacts for the **Zn_24_** complex.

The main intermolecular interaction of **Zn_24_** is H···H contact, which is reflected in the middle of the scattered points of the 2-D fingerprint plots (the percentage of H···H contacts of **Zn_24_** is 40.4%). Another main intermolecular interaction of **Zn_24_** is O···H interaction, which is represented by double spikes in the bottom left (acceptor and donor) region of the fingerprint plots. Accordingly, we can infer that there are significant N-H···O hydrogen bonds (Appendix A) observed in **Zn_24_** (the percentage of O···H contacts of **Zn_24_** is 22.4%). Also, the N···H contacts play important roles for **Zn_24_**. The percentage of N···H contacts of **Zn_24_** is 10.0%. In addition to those above, the presence of C···H, C···O, and N···O contacts were also observed. These three forces accounted for 4.0%, 1.4% and 1.3% of the total Hirshfeld surface force, respectively.

## 4. Conclusions

A supracluster zinc, **Zn_24,_** and a zinc supracluster 1D chain, **1-D⊂Zn_24,_** were synthesized through regulatory solvothermal reactions. In an ethanol solution, **Zn_24_** was synthesized at 353 K using a solvothermal method, whereas **1-D⊂Zn_24_** was formed when a mixed solution (2 mL acetonitrile + 8 mL ethanol) was used at 343 K using a micro bottle. The **Zn_24_** supracluster can be used to selectively detect Ce^3+^ with excellent efficiency (LOD = 8.51 × 10^−7^ mol/L), and can be used as a potential sensor for their detection. In addition, the **Zn_24_** supracluster can detect wavelengths between 302 and 332 nm, using the intensity of emitted light. Thus, the **Zn_24_** supracluster can potentially be used as a spectral detection material to prepare optical wave detectors.

## Data Availability

The data presented in this study are available in this article and Appendix A.

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
