# Peer review of "Zero- to One-Dimensional Zn24 Supraclusters: Synthesis, Structures and Detection Wavelength"

_nanomaterials, 2023, doi:10.3390/nano13233058_

Round 1
Reviewer 1 Report
Comments and Suggestions for Authors
This manuscript can be accepted after following modifications during revision process.
1. It reports the synthesis and structural study of Zn- cluster, the role of secondary interactions are not described properly. Explain it with Hirshfeld surface analysis.
2. Thermal stabilities can be added with TGA and DSC analysis.
3. Phase purity of the materials should be reported with PXRD comparison plots.
4. Morphology of solid materials can be discussed.
5. Why your material selectively detect Ce-ion? Discuss mechanism?
6. NMR of ligands need to add in SI.
Comments on the Quality of English LanguageNeed to revise language.
Author Response
Dear PROF.
Thank you for your time and effort for my submitted nanomaterials No.2727473. according youe suggestion, the paper revised as follows:
Quality of English Language
( ) I am not qualified to assess the quality of English in this paper
( ) English very difficult to understand/incomprehensible
(x) Extensive editing of English language required
( ) Moderate editing of English language required
( ) Minor editing of English language required
( ) English language fine. No issues detected
Response: Thank you for your positive suggestion. The full text of the paper was polished by LetPub Company.

- It reports the synthesis and structural study of Zn- cluster, the role of secondary interactions are not described properly. Explain it with Hirshfeld surface analysis.
Response: Thank you for your positive suggestion. We accepted the editor’s advice and revised in the manuscript and added the Hirshfeld surface analysis.
- Thermal stabilities can be added with TGA and DSC analysis.
Response: Thank you for your positive suggestion. Because the We accepted the editor’s advice and revised in the manuscript.ce the students who synthesized the compound have graduated, we are very sorry to not get enough samples for TG and DSC experiments in such a short time.
- Phase purity of the materials should be reported with PXRD comparison plots.
Response: Thank you for your positive suggestion. We accepted the editor’s advice and added to PXRD as figure S7.
- Morphology of solid materials can be discussed.
Response:Thank you for your positive suggestion. We accepted the editor’s advice and revised in the manuscript.
- Why your material selectively detect Ce-ion? Discuss mechanism?
Response: Thank you for your positive suggestion. Because the addition of Ce3+ can lead to complete quenching in ethanol solution compared with other metal ions. Because the fluorescence quenching mechanism of rare earth ions is very complicated, the fluorescence quenching mechanism is not discussed in this paper.
- NMR of ligands need to add in SI.
Response:Thank you for your positive suggestion. We accepted the editor’s advice and added the 1H NMR for all ligands.
The other revision please see the red part of the paper.
With regards
shuhua zhang
Nov. 26, 2023
Reviewer 2 Report
Comments and Suggestions for Authors
The paper by Chen et al. present the really fascinating result on the synthesis and structural characterisation of the new systems of the molecular wheel type and the 1D supramolecular chains constructed of them. The luminescence properties are also investigated. The paper is of importance and shall be published. However, there is a contrast between the novelty, soundness and interest to the reader on the one hand and the quality of the presentation on the other.
It really takes a while for the reader to understand how the systems were synthesized. Clearly, the procedure of converting a ligand of interest to its Schiff base derivative that is in course of the reaction releases the ligand back is hardly a standard one. Therefore it might be instructive to inform the reader that the reaction of the ATZ with Zn+2 yields an undefined one precipitate. In other words, I would suggest to include the synthetic details with the Scheme 2 before the details of the synthesis. The lack of succes of the direct reaction has to be given as a starting point. Incidentally, concerning Scheme 2, the English word is "precipitate" not "prepipites".
Then the reader knows what is happening and can follow the details of the synthetic procedure. Here is one point that is a bit confusing to me.
The authors begin with the synthesis of " L1H2–L5H2" (line 59). Then, they use the term "H2L1". In Scheme 2, however, we see "H2L2-H2L5" and "H2L1". The first one is a derivative on the unsubstituted salicylic acid as given in the Scheme 2. I would be easier if the authors mentioned it. But what is the difference between L1H2–L5H2 and H2L1 ?
Thus I would suggest the authors the extensive reformulation of the synthetic part. Finally a short comment on the effect of using the particular Br-substituted aldehyde derivative may be of interest.
Author Response
Dear PROF.
Thank you for your time and effort for my submitted nanomaterials No.2727473. according youe suggestion, the paper revised as follows:
Reviewer 2.
The paper by Chen et al. present the really fascinating result on the synthesis and structural characterisation of the new systems of the molecular wheel type and the 1D supramolecular chains constructed of them. The luminescence properties are also investigated. The paper is of importance and shall be published. However, there is a contrast between the novelty, soundness and interest to the reader on the one hand and the quality of the presentation on the other.
Response: Thank you for your positive suggestion.
- It really takes a while for the reader to understand how the systems were synthesized. Clearly, the procedure of converting a ligand of interest to its Schiff base derivative that is in course of the reaction releases the ligand back is hardly a standard one. Therefore it might be instructive to inform the reader that the reaction of the ATZ with Zn+2 yields an undefined one precipitate. In other words, I would suggest to include the synthetic details with the Scheme 2 before the details of the synthesis. The lack of succes of the direct reaction has to be given as a starting point.
Response: Thank you for your positive suggestion. We believe that with the help of Scheme 2, the reader should easily understand the synthesis process. The reaction of the ATZ with Zn2+ yields an undefined one precipitate which may be an interesting compound, but we did not obtain a single crystal, we did not continue to study, sorry.
- Incidentally, concerning Scheme 2, the English word is "precipitate" not "prepipites".
Response: Thank you for your positive suggestion. We have revised the word ‘prepipites’ as ‘precipitates’.
The other revision please see the red part of the paper.
With regards
shuhua zhang
Nov. 26, 2023

Round 2
Reviewer 1 Report
Comments and Suggestions for Authors
The manuscript can be accepted after the clarification of following points.
Why your material selectively detects Ce-ion? Discuss mechanism?
From fig S7. the phase purity is not clear. please explain it.
Comments on the Quality of English LanguageMinor english editing is necessary
Author Response
Dear PROF.
Thank you for your time and effort for my paper. according to your suggestion, the paper revis as follows:
- (x) Minor editing of English language required
Response: we have carefully checked the english language and revised the grammar and spelling errors.
- Why your material selectively detects Ce-ion? Discuss mechanism?
Response: Thank you for your positive suggestion. We studied most rare earth ions (Figure 5) and found that only Ce-ion had a quenching effect, so we chose to detect cerium ions. Because the fluorescence quenching mechanism of rare earth ions is very complicated, the fluorescence quenching mechanism is not discussed in this paper. Thank you so much for agreeing with us to do so.
- From fig S7. the phase purity is not clear. please explain it.
Response: We believe that the peak sites of most of the powder diffraction curve can match most of the peak positions of the simulated powder diffraction curve, indicating that the purity of the sample is ok. However, due to the presence of very large pores in this compound, some of the solvent molecules are not resolved, so it is normal to have some differences between the simulation curve and the experimental curve, very sorry.
The other revision please see the red part of the paper.
With regards
shuhua zhang
Nov. 27, 2023